# The effects of omega-3 fatty acids on diabetic nephropathy: A meta-analysis of randomized controlled trials

Api Chewcharat[1,2]*, Pol Chewcharat[3], Anawin Rutirapong[3], Stefania Papatheodorou[1]

1 Department of Epidemiology, Harvard T.H. Chan School of Public Health, Boston, MA, United States of America, 2 Division of Nephrology and Hypertension, Mayo Clinic, Rochester, MN, United States of America, 3 Department of Internal Medicine, Faculty of Medicine, Chulalongkorn University, Bangkok, Thailand

* Api.che@hotmail.com

## Abstract

### Objective

To evaluate the effects of omega-3 long-chain polyunsaturated fatty acids on proteinuria, estimated glomerular filtration rate (eGFR) and metabolic biomarkers among patients with diabetes.

### Study design

Meta-analysis of randomized controlled clinical trials (RCTs).

### Setting & subjects

Patients with diabetes.

### Selection criteria for studies

We conducted electronic searches in PubMed, Embase and Cochrane Central Register of Controlled Trials from January 1960 to April 2019 to identify RCTs, which examined the effects of omega-3 fatty acids on proteinuria, eGFR and metabolic biomarkers among diabetic patients.

### Results

Ten RCTs with 344 participants were included in our meta-analysis. Omega-3 fatty acids reduced the amount of proteinuria among type 2 diabetes mellitus (type 2 DM) and type 1 diabetes mellitus (type 1 DM). This association was only significant among type 2 DM (SMD = -0.29 (95% CI: -0.54, -0.03; p = 0.03). Only studies with duration of intervention of 24 weeks or longer demonstrated a significant lower proteinuria among omega-3 fatty acids compared to control group (SMD = -0.30 (95% CI: -0.58, -0.02; p = 0.04). There was a higher eGFR for both type 1 and type 2 DM groups among omega-3 fatty acids compared to control group, however, the effect was not statistically significant. Regarding serum total cholesterol, LDL-cholesterol and HbA1C, there was no significant difference comparing

**Data Availability Statement:** All relevant data are within the manuscript and its Supporting Information files.

**Funding:** The authors received no specific funding for this work

**Competing interests:** No authors have competing interests.

omega-3 fatty acids to control group. There was a non-significant systolic blood pressure reduction in the omega-3 fatty acids supplementation group compared to control.

## Conclusion

Omega-3 fatty acids could help ameliorate proteinuria among type 2 DM who received omega-3 supplementation for at least 24 weeks without adverse effects on HbA1C, total serum cholesterol and LDL-cholesterol.

## Introduction

The prevalence of diabetes around the world has reached an unprecedented level in recent decades. While diabetes is already estimated to afflict more than 350 million people around the world, this is predicted to grow to over 550 million people by the year 2035[1, 2]. More importantly, 30–40% of patients with diabetes mellitus will develop diabetic nephropathy[2] which is characterized by proteinuria in advanced stages. The degree of proteinuria reflects the severity of glomerular damage and is associated with a faster decline in the estimated glomerular filtration rate (eGFR) [3–5]. Additionally, proteinuria in this population is associated with hyperuricemia, stroke, and cardiovascular disease morbidity/mortality [5–8].

Long-chain omega-3 polyunsaturated fatty acids, including eicosapentaenoic acid (EPA) and docosahexaenoic acids (DHA), have shown anti-inflammatory, antithrombotic properties and benefits on kidney function[9–11]. There is a number of clinical trials studying in various types of kidney diseases including IgA nephropathy[12], lupus nephritis[13, 14] and polycystic kidney disease[15]. However, the information about the effects of omega-3 fatty acids on kidney function, particularly in diabetic kidney disease still lacks consensus. Currently, the data from Diabetes Control and Complications Trial showed that higher dietary eicosapentaenoic acid and docosahexaenoic acid consumption was associated with a lower risk of proteinuria among diabetic patients[16]. Nonetheless, the meta-analysis on the effect of n–3 long-chain polyunsaturated fatty acid supplementation on urine protein excretion and kidney function by Miller et al.[17] in 2009 suggested that there was no sufficient evidence to conclude that n–3 long-chain polyunsaturated fatty acid supplementation could reduce albuminuria among diabetic patients subgroup (7 studies, 222 patients). Since then, 3 new studies were published including 344 patients (55% increases in sample size). Moreover, another meta-analysis on omega-3 fatty acid supplementation as adjunctive therapy in the treatment of chronic kidney disease by Jing et al.[11] in 2017 suggested that omega-3 fatty acid supplementation is associated with a significantly reduced risk of end-stage renal disease and delays the progression of this disease, but in this study, diabetic patients were not included.

The aim of this meta-analysis was to investigate the effects of omega-3 fatty acid supplementation in reducing proteinuria in diabetic patients by using all available evidence from the published literature. All eligible studies assessed proteinuria, the serum creatinine clearance rate, the estimated glomerular filtration rate, or the occurrence of end-stage renal disease.

## Methods

### Data sources and searches

The protocol for this systematic review is registered with PROSPERO (International Prospective Register of Systematic Reviews; no.CRD42019134873). We conducted electronic searches

in PubMed, Embase and Cochrane Central Register of Controlled Trials from January 1960 to April 2019 to identify randomized controlled trials (RCTs), which explored the effects of omega-3 fatty acid supplementation on proteinuria, eGFR and metabolic biomarkers among diabetic patients. The same search strategy was used for EMBASE and Cochrane Central Register of Controlled Trials using the corresponding terms. Manual searches of the reference lists from all relevant original and review articles were also conducted to identify additional eligible studies. This study was conducted by the Preferred Reporting Items for Systematic Reviews and Meta-Analysis (PRISMA) statement[18].

## Selection criteria

RCTs examining the effect of omega-3 fatty acid supplementation compared to control on proteinuria or albuminuria were included. There were no restrictions on sample size or study duration. Retrieved articles were individually reviewed for eligibility by two investigators (A.C. and A.R.). Disagreements were addressed and solved by mutual consensus.

## Data extraction and quality assessment

The following data were extracted: study design, year of publication, country of origin, sample size, duration of follow-up, type of omega-3 fatty acid, dose, frequency, mean age and type of diabetes. The following outcomes of interest were examined: change in kidney outcomes [proteinuria and eGFR], serum lipids and glucose control biomarkers [triglyceride, total cholesterol (TC), high density lipoprotein (HDL), low density lipoprotein (LDL), hemoglobin A1C (HbA1C)] and blood pressure parameters [systolic blood pressure (SBP), diastolic blood pressure (DBP)] between baseline and at the study end.

Revised Cochrane risk-of-bias tool for randomized trials (RoB 2)[19] was used to evaluate the risk of bias for RCTs. The assessment included the following components: risk of bias arising from randomization process, risk of bias due to deviation from the intended interventions, missing outcome data, risk of bias in measurement of the outcome and risk of bias in the selection of the reported result. A judgment about the risk of bias arising from each domain is generated by an algorithm, based on answers to the signaling questions. Judgment could be high risk of bias, low risk of bias, or some concerns.

## Data synthesis and statistical analysis

Random effects models were used due to the expected clinical heterogeneity in the included populations. We also compared the results with the fixed effect model. Adjusted point estimates from each study were consolidated by the generic inverse variance approach of DerSimonian and Laird, which designated the weight of each study based on its variance[20]. We also applied fixed effects models to compare the results. We computed standardized mean difference (SMD) in mean values for proteinuria at the study end because this particular outcome was measured on a different scale across studies. However, for other continuous variables that were measured on the same scale, we used weight mean difference (WMD) for the mean values at the study end. We assumed that there were no significant differences in baseline characteristics for each variable in randomized controlled trials. All pooled estimates were displayed with 95% confidence intervals (CI). Heterogeneity among effect sizes estimated by individual studies was described with the $I^2$ index and the chi-square test. A value of $I^2$ of 0%-25% represents insignificant heterogeneity, 26%-50% low heterogeneity, 51%-75% moderate heterogeneity and 76–100% high heterogeneity[21]. Meta-regression was used to assess the association between change in proteinuria and change in eGFR as well as the change in proteinuria and combined dose of DHA and EPA.

Publication bias was formally assessed using funnel plots and the Egger test to assess for asymmetry of the funnel plot. A p-value of less than 0.05 indicates the presence of publication bias[22]. The meta-analysis was performed by STATA/IC 15.1 (StataCorp LLC, Texas, USA).

# Results

## Characteristics and quality of the studies

A total of 1,277 potentially relevant citations were identified and screened. Seventy citations were evaluated in detail, of which 10 trials [23–32] with 344 participants fulfilled the eligibility criteria and were included in this meta-analysis. The literature retrieval, review, and selection process are demonstrated in Fig 1.

Characteristics of the individual trials are displayed in Table 1. Briefly, the trials varied in sample size from 18 to 79 patients. From 10 trials, three followed a cross-over design[24, 25, 27]. There were 3 trials conducted in North America [23–25], 3 trials conducted in Europe [26, 27, 29], 3 trials conducted in Asia [28, 31, 32] and 1 trial conducted in Australia [30]. There were 5 trials that included only type 2 DM [23, 24, 27, 28, 32], 3 trials that included only type 1 DM [25, 26, 29] and 2 trials that included both type 1 and type 2 DM [30, 31]. However, only one from those two studies reported outcomes in each group separately[31]. For the

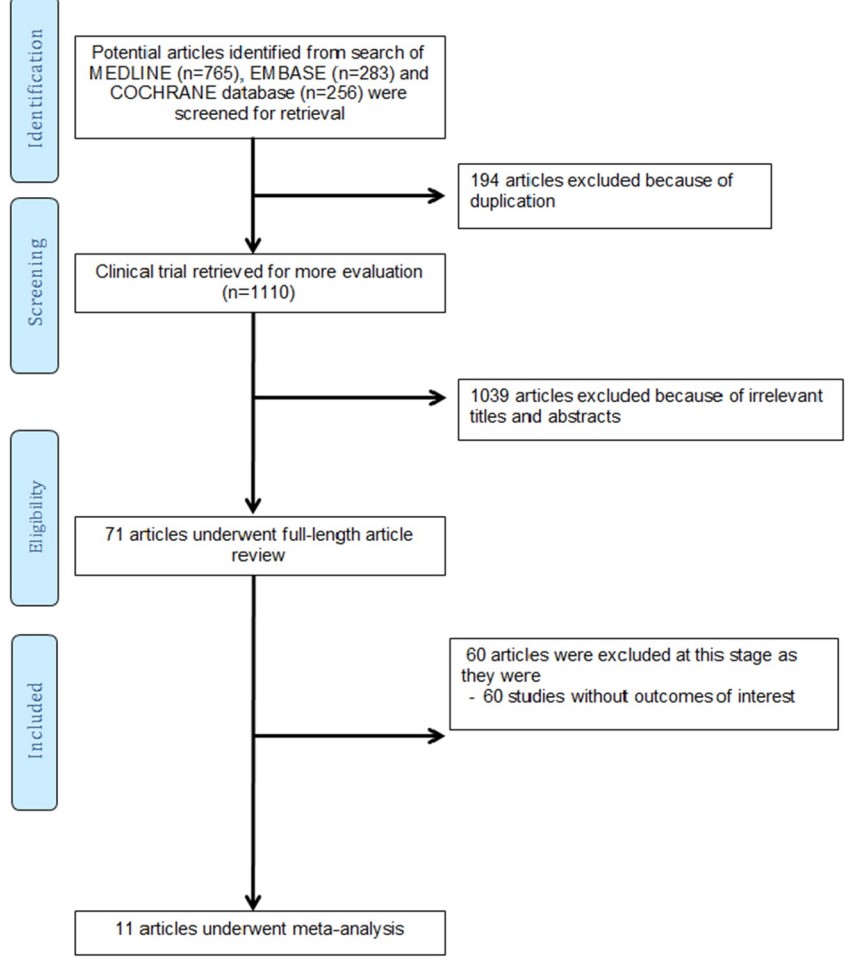

**Fig 1. Search methodology and selection process.**

**Table 1. Main characteristics of studies included in the meta-analysis of the effects of omega-3 fatty acids on proteinuria among patients with diabetes.**

| Author | Country | Number of patients | Mean age | % Female | Baseline eGFR (ml/min/ 1.73 m$^2$) | EPA | DHA | Control | Follow-up | Type of DM |
|---|---|---|---|---|---|---|---|---|---|---|
| | | | | | | Fish oil | | Control | | |
| Haines 1986 [26] | UK | 41 | 42.3 | 26.8 | NA | 2.7 g | 1.9 g | olive oil | 6 wk | Type 1 |
| Jensen 1989 [25] | USA | 18 | 37.0 | 22.2 | 82±5 | 2 g | 2.6 g | olive oil | 8 wk | Type 1 |
| Hamazaki 1990 [31] | Japan | 9 | 59.3 | 55.6 | NA | 1.8 g | - | no omega-3 supplementation | 24 wk | Type 1 |
| | | 17 | 66.0 | 58.8 | NA | 1.8 g | - | no omega-3 supplementation | 24 wk | Type 2 |
| Shimizu 1995 [28] | Japan | 45 | 63.6 | 51.2 | NA | 900 mg | - | healthy | 52 wk | Type 2 |
| Rossing 1996 [29] | Denmark | 29 | 33.0 | 34.5 | 116±7 | 2.0 g | 2.6 g | olive oil | 52 wk | Type 1 |
| Lungershausen 1997 [30] | Australia | 32 | 55.0 | 25.0 | 116±11 | 2.0 g | 1.4 g | corn oil | 12 wk | Type 1 and type 2 |
| Zeman 2005 [27] | Czech | 24 | 48.8 | 45.8 | NA | 2.07 g | 1.53 g | olive oil | 52 wk | Type 2 |
| Miller 2013 [24] | USA | 31 | 67.4 | 45.1 | 78±22 | 2.26 g | 1.13 g | placebo | 8 wk | Type 2 |
| Lee 2015 [32] | Korea | 19 | 60.4 | 36.8 | 58±8 | 1.38 g | 1.14 g | olive oil | 12 wk | Type 2 |
| Elajami 2017 [23] | USA | 79 | 63.4 | 18.9 | 79±22 | 1.86 g | 1.5 g | no omega-3 supplementation | 52 wk | Type 2 |

analysis purposes, the study by Hamazaki et al. was divided into 2 separate studies based on the type of DM. Therefore, we had 11 study arms from 10 original studies and no duplicate populations. The mean age of patients ranged from 33 to 67.4 years old. The duration of follow up spanned from 6 weeks to 52 weeks.

## Risk of bias

According to the revised Cochrane risk-of-bias tool for randomized trials, with respect to the overall risk of bias, five studies had low risk of bias [24, 25, 29, 30, 32]; one study with some concerns for risk of bias [23] and another four studies had high risk of bias [26–28, 31]. In terms of risk of bias arising from the randomization process, four studies had high risk of bias [26–28, 31]. For risk of bias due to deviations from the intended interventions, five studies raised some concerns [23, 26–28, 31]. Five studies raised some concerns for missing outcome data and risk of bias in selection of the reported result [23, 26–28, 31]. All of the studies had low risk of bias in the measurement of the outcome. There were five studies that had some concerns for the risk of bias in selection of the reported result [23, 26–28, 31]. There was no study that had high risk of bias in all domains (Table 2).

## Effect of omega-3 fatty acids on kidney outcomes

As shown in Fig 2 and Table 3, 11 study arms (342 patients) reported proteinuria as the primary outcome. We found that proteinuria among diabetic patients receiving omega-3 fatty acids was lower than control group (SMD = -0.19 (95% CI: -0.38, 0.01); p = 0.06, $I^2$ = 0%) but this was not statistically significant. Six study arms (208 patients) showed a higher eGFR among omega-3 fatty acids group but the effect was not significant (WMD = 1.56 mL/min/ 1.73m$^2$ (95% CI:-1.53, 4.65); p = 0.32, $I^2$ = 5.6%).

**Table 2. Risk of bias according to revised Cochrane risk-of-bias tool for randomized trials.**

| | Risk of bias arising from the randomization process | Risk of bias due to deviations from the intended interventions | Missing outcome data | Risk of bias in measurement of the outcome | Risk of bias in selection of the reported result | Overall risk of bias |
|---|---|---|---|---|---|---|
| Haines 1986 [26] | High | Some concerns | Some concerns | Low | Some concerns | high |
| Jensen 1989 [25] | Low | Low | Low | Low | Low | Low |
| Hamazaki 1990 [31] | High | Some concerns | Some concerns | Low | Some concerns | high |
| Shimizu 1995 [27] | High | Some concerns | Some concerns | Low | Some concerns | high |
| Rossing 1996 [29] | Low | Low | Low | Low | Low | Low |
| Lungershausen 1997 [30] | Low | Low | Low | Low | Low | Low |
| Zeman 2005 [27] | High | Some concerns | Some concerns | Low | Some concerns | high |
| Miller 2013 [24] | Low | Low | Low | Low | Low | Low |
| Lee 2015 [32] | Low | Low | Low | Low | Low | Low |
| Elajami 2017 [23] | Low | Some concerns | Some concerns | Low | Some concerns | Some concerns |

## Effect of omega-3 fatty acids on blood pressure parameters

Ten study arms with 318 patients reported that there were no differences in both SBP (WMD = -2.10 mmHg (95% CI:-4.48, 0.28); P = 0.08, $I^2$ = 0%), and DBP (WMD = 1.04 mmHg (95% CI:-1.81, 3.89); P = 0.48, $I^2$ = 39.8%) between treatment group and control group as shown in Figs 3A and 3B.

## Effect of omega-3 fatty acids on serum lipids and glucose control

Regarding triglycerides, ten study arms with 313 patients showed that omega-3 fatty acids significantly diminished triglycerides (WMD = -24.24 mg/dL (95% CI:-36.40, -12.10); P < 0.001, $I^2$ = 0%). While in lights of total cholesterol, six study arms with 168 participants demonstrated no significant difference for total cholestrol between omega-3 fatty acids group and control (WMD = 3.72 mg/dl (95% CI:-4.63, 12.06); P = 0.38, $I^2$ = 80.2%). In terms of serum LDL-cholesterol, six study arms with 215 patients demonstrated no significant difference in serum LDL-cholesterol (WMD = 2.29 mg/dL (95% CI:-2.45, 7.03); P = 0.34, $I^2$ = 0%). However, for HDL-cholesterol, six study arms with 242 participants illustrated that omega-3 fatty acids group had a higher HDL-cholesterol compared to control group (WMD = 4.57 mg/dL (95% CI: 0.79, 8.34); P = 0.02, $I^2$ = 82.5%). Moreover, ten study arms with 313 patients illustrated no significant difference in HbA1C between omega-3 fatty acids group and control group (WMD = -0.03% (95% CI: -0.45, 0.39); P = 0.89, $I^2$ = 66.2%). Forrest plots were shown in S1–S6 Figs.

## Fixed effects models

We also performed the analyses using fixed effects models. DBP, total cholesterol and HbA1C became significantly different between omega-3 fatty acids and control group as shown in S1 Table. However, a random effects model will yield more conservative results than the fixed effect when $tau^2$ is not equal to zero.

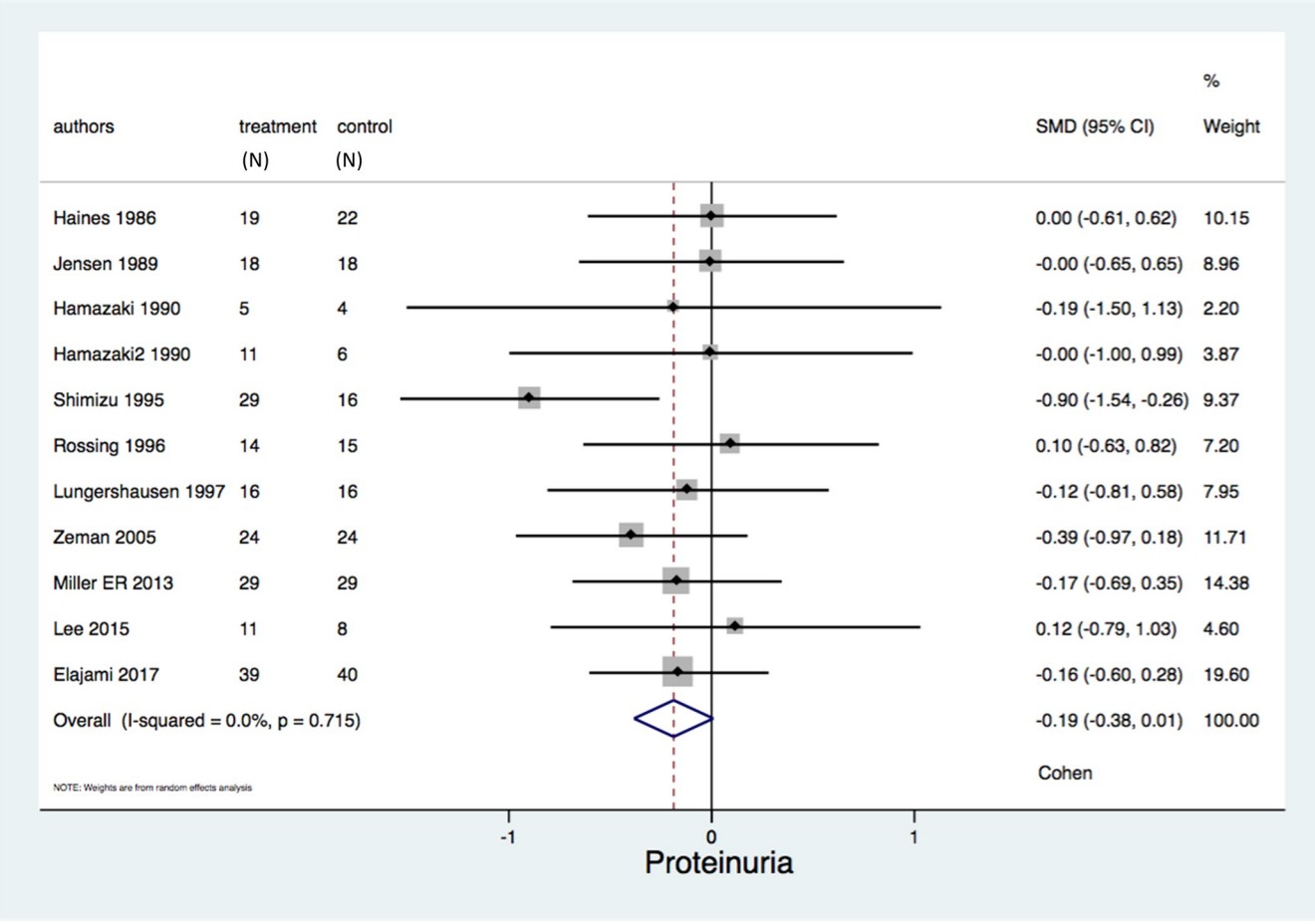

Hamazaki2 is the study by Hamazaki et al. that mainly focused on type 2 DM group

**Fig 2. Forest plots of the included studies assessing proteinuria among diabetic patients.**

**Table 3. Summary effects of omega-3 fatty acids on outcomes of interest among diabetic patients.**

| Outcomes | No of study arms | No of patients | Weighted mean difference/Standardized mean difference* | Confidence interval | $I^2$ | P-value |
|---|---|---|---|---|---|---|
| Proteinuria | 11 | 342 | -0.19* | (-0.38, 0.01) | 0% | 0.06 |
| eGFR | 6 | 208 | 1.56 mL/min/1.73 m² | (-1.53, 4.65) | 5.6% | 0.32 |
| SBP | 10 | 318 | -2.10 mmHg | (-4.48, 0.28) | 0% | 0.08 |
| DBP | 10 | 318 | 1.04 mmHg | (-1.81, 3.89) | 39.8% | 0.48 |
| Triglyceride | 10 | 313 | -24.24 mg/dL | (-36.40, -12.10) | 0% | **<0.001** |
| TC | 6 | 168 | 3.72 mg/dl | (-4.63, 12.06) | 80.2% | 0.38 |
| HDL-c | 6 | 242 | 4.57 mg/dl | (0.79, 8.34) | 82.5% | **0.02** |
| LDL-c | 6 | 215 | 2.29 mg/dL | (-2.45, 7.03) | 0% | 0.34 |
| HbA1C | 10 | 313 | -0.03% | (-0.45, 0.39) | 66.2% | 0.89 |

eGFR, estimated glomerular filtration rate; TC, total cholesterol; LDL-c, low density lipoprotein cholesterol; HDL-c, high density lipoprotein cholesterol; HbA1C, hemoglobin A1C; SBP, systolic blood pressure; DBP, diastolic blood pressure

* indicates standardized mean differences

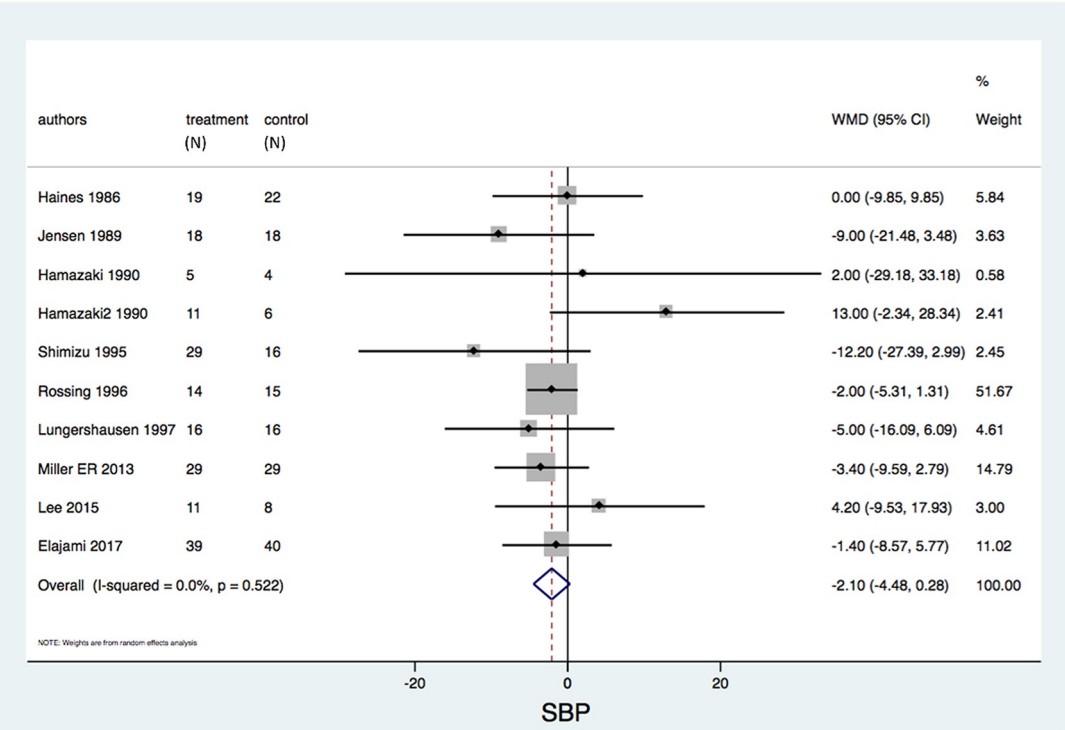

Hamazaki2 is the study by Hamazaki et al. that mainly focused on type 2 DM group

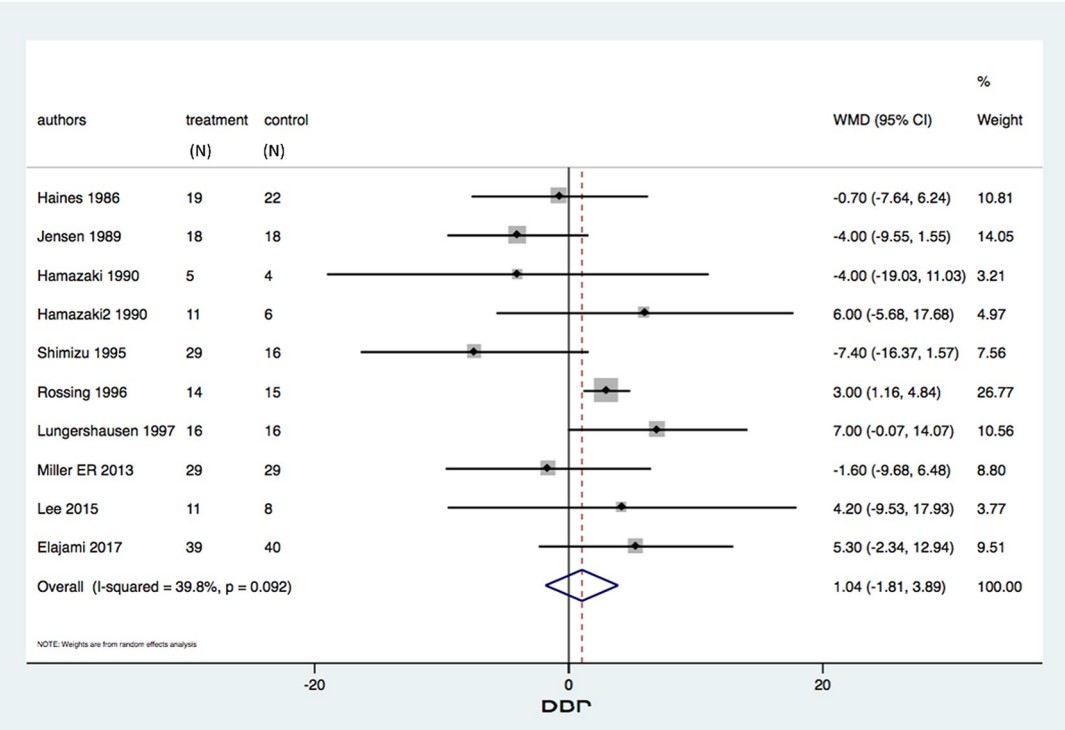

Hamazaki2 is the study by Hamazaki et al. that mainly focused on type 2 DM group

**Fig 3.** a Forest plots of the included studies assessing systolic blood pressure among diabetic patients. b Forest plots of the included studies assessing diastolic blood pressure among diabetic patients.

## Subgroup analysis and meta-regression

In the subgroup analysis for type of DM, we excluded the study by Lungershausen et al.[30] since they did not provide separate results according to type of DM. Among type 2 DM group with 213 participants, omega-3 fatty acids could significantly reduce proteinuria (SMD = -0.29 (95% CI: -0.54, -0.03); P = 0.03, $I^2$ = 3.9%) when compared to control group. However, among type 1 DM group with 97 participants, there was no significant difference in proteinuria (SMD = 0.01 (95% CI -0.36, 0.38); P = 0.95, $I^2$ = 0%) between omega-3 fatty acids group and control group (Fig 4). For serum triglyceride, lower serum triglyceride was found among omega-3 fatty acids group in both type 1 diabetes with 97 participants (WMD = -29.35 mg/dl (-55.53, -3.18); p-value = 0.03, $I^2$ = 0%) and type 2 diabetes with 213 participants (WMD = -21.36 mg/dl (-39.24, -3.47); p-value = 0.02, $I^2$ = 32.1%). However, for HDL cholesterol, 70 participants with type 1 diabetes demonstrated a higher HDL compared to control group (WMD = 8.07 mg/dl (0.45, 15.70); p-value = 0.04, $I^2$ = 86.1%) while type 2 DM with 172

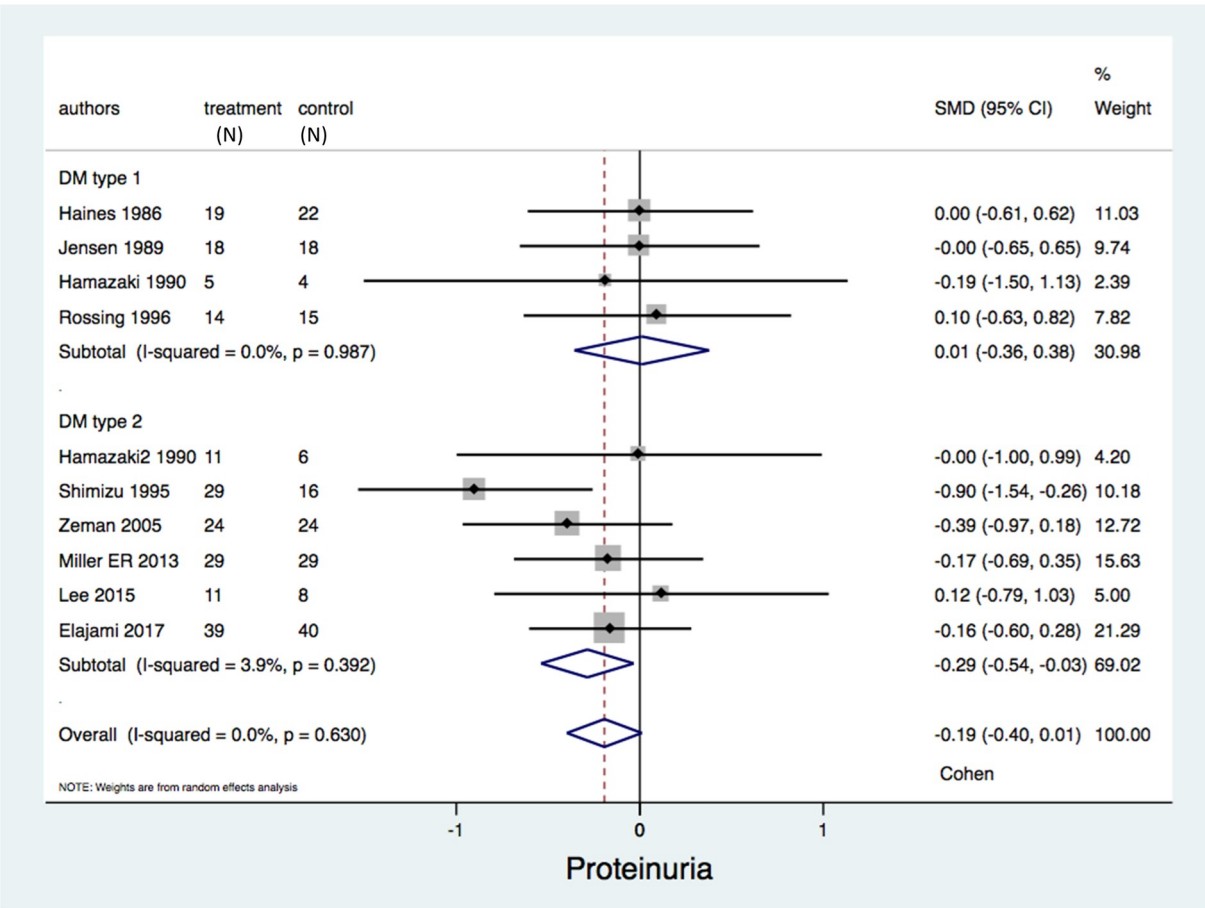

Hamazaki2 is the study by Hamazaki et al. that mainly focused on type 2 DM group

**Fig 4. Forest plots of the included studies assessing proteinuria among diabetic patients categorized by type of diabetes.**

participants failed to reveal significant difference in HDL between omega-3 fatty acids group and control group (WMD = 2.59 mg/dl (-1.40, 6.57); p-value = 0.20, $I^2$ = 67.5%). Other parameters of interest are shown in Table 4.

Stratified by the duration of follow-up, we used 24 weeks as a cut point since this value was a median. We found that study with follow-up time at least 24 weeks (203 participants) demonstrated a significant reduction in proteinuria comparing omega-3 fatty acids to control group (SMD = -0.30 (-0.58, -0.02); p-value = 0.04, $I^2$ = 6.1%) while study with follow-up period less than 24 weeks (139 participants) failed to show significant difference in proteinuria (SMD = -0.06 (-0.35, 0.23); p-value = 0.68, $I^2$ = 0%). Other parameters were shown in Table 4.

Moreover, we found that only type 2 DM patients who received omega-3 fatty acids for at least 24 weeks (165 participants) had a significant decrease in proteinuria comparing to control group (SMD = -0.38 (-0.73, -0.03); p-value = 0.04, $I^2$ = 24.8%). While among type 1 DM patients, there was no significant difference in decreasing proteinuria even supplementing with omega-3 fatty acids for more than 24 weeks (38 participants) (SMD = 0.03 (-0.61, 0.67); p-value = 0.93, $I^2$ = 0%). In a meta-regression analysis, the change in proteinuria was not associated with change in GFR (-0.01 (-0.09, 0.07); p-value = 0.69) and the change in proteinuria was not associated with combined dose of EPA and DHA (0.03 (-0.17, 0.24); p-value = 0.73).

### Assessment of publication bias

As Egger's test for proteinuria as our primary outcome was not significant (P > 0.05), together with a funnel plot for proteinuria of the studies included in this meta-analysis without significant asymmetry. Therefore, publication bias was less likely to occur. (Fig 5)

### Discussion

Even though several meta-analyses have previously investigated the effects of omega-3 fatty acids on proteinuria, the possible benefits of omega-3 fatty acids remain unclear, especially among diabetic patients. This is the largest meta-analysis to assess the treatment effect of omega-3 fatty acids on proteinuria and other outcomes among different types of diabetic patients. Our meta-analysis demonstrated that omega-3 fatty acids could ameliorate proteinuria among type 2 DM who received this supplementation for at least 24 weeks. However, there were no significant effects on eGFR, serum LDL-cholesterol, serum HbA1C and blood pressure parameters. We included 344 patients with both type 2 DM and type 1 DM in RCTs from 1960 to April 2019. A previous meta-analysis by Miller et al.[17] in 2009 included only 222 diabetic patients, which suggested insufficient data to confirm the efficacy of omega-3 fatty acid treatments for proteinuria in diabetic patients. Moreover, we also performed subgroup analysis in terms of type of diabetes and follow-up period to gain more insight on the exploration of heterogeneity and we found a significant effect of omega-3 fatty acids on reducing proteinuria among type 2 DM and among patients with a follow-up period of at least 24 weeks.

The mechanisms through which omega-3 fatty acids diminish proteinuria are not clear. Evidence suggests that omega-3 fatty acids may act via renal hemodynamic effects[33]. However, in our meta-analysis, the observed effects of omega-3 fatty acids supplementation on proteinuria are not likely the result of blood pressure or renal perfusion effects because we did not observe any significant differences in blood pressure parameters.The effect of omega-3 fatty acids in ameliorating proteinuria may be beyond hemodynamic parameters. One of the hypotheses is that omega–3 fatty acids may reduce urine protein excretion through anti-inflammatory effects and oxidative stress. As hyperglycemia among diabetic patients induces podocyte injury as well as endothelial cell and tubulointerstitial injury through the formation

**Table 4. Summary effects of subgroup analysis on the type of diabetes and follow-up period on omega-3 fatty acids on outcomes of interest among diabetic patients.**

| Outcomes | Mean difference | 95% CI | P-value | I² |
|---|---|---|---|---|
| Proteinuria | SMD | | | |
| Type of diabetes | | | | |
| Type 2 | -0.29 | (-0.54, -0.03) | **0.03** | 3.9% |
| Type 1 | 0.01 | (-0.36, 0.38) | 0.95 | 0% |
| Follow-up period | | | | |
| < 24 weeks | -0.06 | (-0.35, 0.23) | 0.68 | 0% |
| > = 24 weeks | -0.30 | (-0.58, -0.02) | **0.04** | 6.1% |
| eGFR | WMD | | | |
| Type of diabetes | | | | |
| Type 2 | 1.34 mL/min/1.73m² | (-4.94, 7.61) | 0.68 | 59.2% |
| Type 1 | 1.88 mL/min/1.73m² | (-2.90, 6.67) | 0.44 | 0% |
| Follow-up period | | | | |
| < 24 weeks | -0.70 mL/min/1.73m² | (-4.80, 3.40) | 0.74 | 0% |
| > = 24 weeks | 4.35 mL/min/1.73m² | (-1.39, 10.09) | 0.14 | 40.6% |
| SBP | WMD | | | |
| Type of diabetes | | | | |
| Type 2 | -0.95 mmHg | (-6.69, 4.79) | 0.75 | 37.1% |
| Type 1 | -2.19 mmHg | (-5.21, 0.84) | 0.16 | 0% |
| Follow-up period | | | | |
| < 24 weeks | -2.93 mmHg | (-7.15, 1.28) | 0.17 | 0% |
| > = 24 weeks | -1.36 mmHg | (-6.08, 3.36) | 0.57 | 26.7% |
| DBP | WMD | | | |
| Type of diabetes | | | | |
| Type 2 | 0.78 mmHg | (-4.40, 5.97) | 0.77 | 31.9% |
| Type 1 | -0.16 mmHg | (-4.44, 4.13) | 0.94 | 55.9% |
| Follow-up period | | | | |
| < 24 weeks | 0.29 mmHg | (-3.98, 4.55) | 0.90 | 36.6% |
| > = 24 weeks | 1.67 mmHg | (-2.51, 5.85) | 0.43 | 38.3% |
| Triglyceride | WMD | | | |
| Type of diabetes | | | | |
| Type 2 | -21.36 mg/dl | (-39.24, -3.47) | **0.02** | 32.1% |
| Type 1 | -29.35 mg/dl | (-55.53, -3.18) | **0.03** | 0% |
| Follow-up period | | | | |
| < 24 weeks | -23.10 mg/dl | (-41.33, -4.87) | **0.01** | 0% |
| > = 24 weeks | -24.60 mg/dl | (-43.99, -5.20) | **0.01** | 14.7% |
| TC | WMD | | | |
| Type of diabetes | | | | |
| Type 2 | 1.96 mg/dl | (-11.36, 15.28) | 0.77 | 86.6% |
| Type 1 | 6.79 mg/dl | (-4.52, 18.10) | 0.24 | 62.2% |
| Follow-up period | | | | |
| < 24 weeks | -0.91 mg/dl | (-8.47, 6.64) | 0.81 | 0% |
| > = 24 weeks | 5.99 mg/dl | (-5.51, 17.48) | 0.31 | 88.2% |
| HDL-c | WMD | | | |
| Type of diabetes | | | | |
| Type 2 | 2.59 mg/dl | (-1.40, 6.57) | 0.20 | 67.5% |
| Type 1 | 8.07 mg/dl | (0.45, 15.70) | **0.04** | 86.1% |

*(Continued)*

**Table 4.** (Continued)

| Outcomes | Mean difference | 95% CI | P-value | $I^2$ |
|---|---|---|---|---|
| Follow-up period | | | | |
| < 24 weeks | 1.53 mg/dl | (-1.43, 4.50) | 0.31 | 0% |
| > = 24 weeks | 6.60 mg/dl | (1.47, 11.72) | **0.01** | 88.0% |
| LDL-c | WMD | | | |
| Type of diabetes | | | | |
| Type 2 | -0.26 mg/dl | (-7.08, 6.56) | 0.94 | 0% |
| Type 1 | 4.67 mg/dl | (-1.92, 11.25) | 0.17 | 0% |
| Follow-up period | | | | |
| < 24 weeks | 2.23 mg/dl | (-5.59, 10.04) | 0.58 | 0% |
| > = 24 weeks | 1.86 mg/dl | (-5.50, 9.23) | 0.62 | 27.8% |
| HbA1C | WMD | | | |
| Type of diabetes | | | | |
| Type 2 | -0.14% | (-0.55, 0.26) | 0.50 | 20.5% |
| Type 1 | 0.27% | (-0.73, 1.27) | 0.60 | 81.9% |
| Follow-up period | | | | |
| < 24 weeks | 0.33% | (-0.18, 0.83) | 0.21 | 0% |
| > = 24 weeks | -0.22% | (-0.73, 0.29) | 0.40 | 69.5% |

eGFR, estimated glomerular filtration rate; TC, total cholesterol; LDL-c, low density lipoprotein cholesterol; HDL-c, high density lipoprotein cholesterol; HbA1C, hemoglobin A1C; SBP, systolic blood pressure; DBP, diastolic blood pressure; WMD, weighted mean differences; SMD, standardized mean difference

of advanced glycation end-products (AGE), activation of protein kinase C (PKC) and generation of reactive oxygen species, this process plays a pivotal role in initiation and progression of proteinuria and diabetic nephropathy[34].

Our meta-analysis demonstrated only the benefits in delaying proteinuria among type 2 DM patients. This could be explained by a small sample size of type 1 DM patients (213 vs 97). Additionally, the pathophysiology of diabetic nephropathy in type 2 DM and type 1 DM patients is somewhat different. For type 2 DM, proteinuria could be caused by various etiologies including but not limited to insulin resistance, concomitant hypertension and obesity. One of the possible explanations would be that among type 2 diabetes there are pro-inflammatory cytokines generated from abundant adipose tissue as a part of obesity in type 2 diabetes. This inflammatory response leads to proteinuria among diabetic nephropathy. Omega-3 fatty acids help reduce insulin resistance as well as pro-inflammatory responses from adipose tissue. This effect might result in lower proteinuria compared to patients with type 1 diabetes which proteinuria is mainly through polyol, hexosamine, advanced glycation end product and protein kinase C (PKC) pathways [35, 36]. Nevertheless, any meta-analyses could not derive explanations for any mechanistic pathways or derive a hypothesis. Hence, future studies designed to examine mechanisms of omega-3 fatty acids on proteinuria or kidney functions are needed as well as to assess the effect of omega-3 fatty acids on inflammatory cytokines among type 1 and type 2 diabetes.

We found that omega-3 fatty acids did not provide any effects on GFR decline. This could be explained by a low sample size as well as short period of follow-up. Furthermore, we knew that there were about one-third of proteinuric patients who did not develop end-stage renal disease (ESRD) after 20 years of follow-up and about 10% of diabetic patients without proteinuria whose kidney function kept declining and led to ESRD[37, 38]. Therefore, proteinuria and GFR decline is loosely correlated as we also found by meta-regression. However,

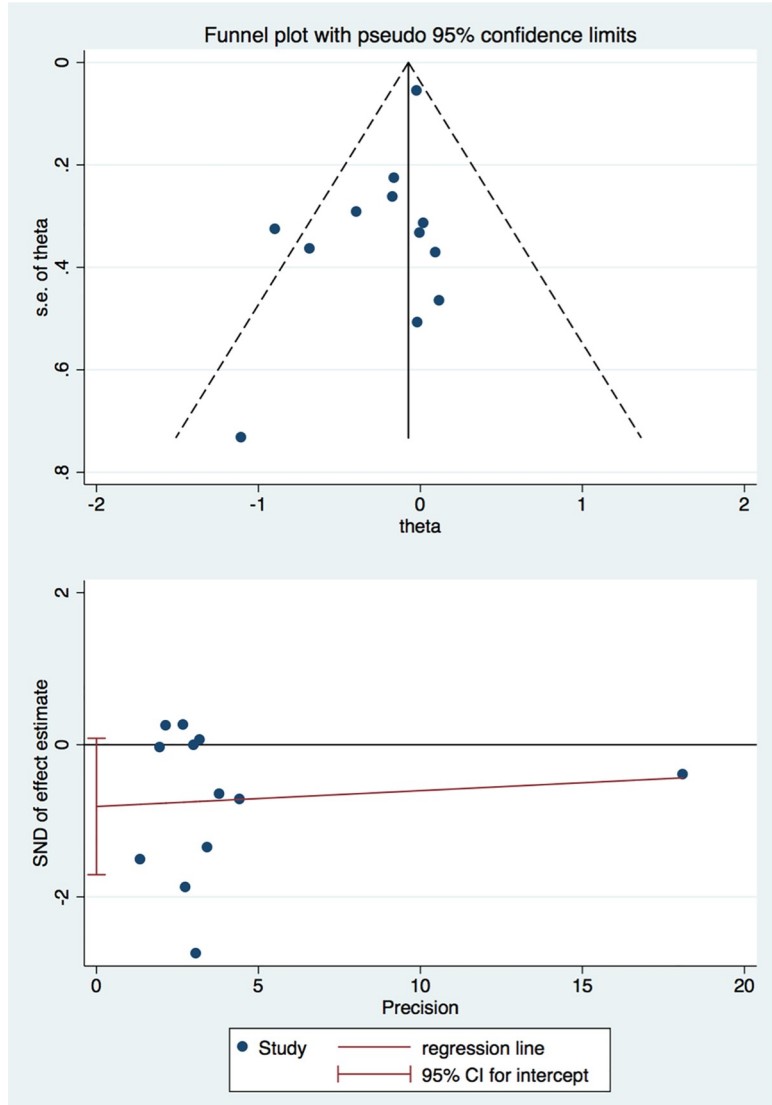

**Fig 5. Funnel plot of standardized mean difference of proteinuria.**

proteinuria is still a predictor of cardiovascular and stroke events among diabetic patients. We hypothesized that omega-3 fatty acids could help diminish proteinuria and reduce cardiovascular complications and stroke incidence among type 2 DM.

In terms of effects on lowering blood pressure of omega-3 fatty acids, our findings are consistent with the previous meta-analysis of the effects of omega-3 acids on cardiometabolic biomarkers in type 2 diabetes by Lauren et al. in 2018 [39] which included 2674 patients. With respect to HbA1C, the effect of omega-3 fatty acids on HbA1C is controversial. A meta-analysis by Zhou et al.[40] found that intake of omega-3 fatty acids might be associated with increased type 2 diabetes risk. It raised the concern that omega-3 fatty acids intake might interfere with HbA1C control. However, our meta-analysis revealed no significant difference in HbA1C between treatment arms and control group which is congruent with the latest meta-analysis on the same topic for HbA1C by Chen et al [41]. Lastly, regarding the effects of omega-3 fatty acids on blood lipid level, it aligns with the previous meta-analysis [39, 42]

which showed a significant reduction in serum triglyceride. However, our meta-analysis did not find a significant reduction in LDL. This might be explained by our small sample size to conclude the effect on serum lipid profile. Additionally, we found that omega-3 fatty acids significantly raised serum HDL only among type 1 diabetes. This could be explained by higher doses of omega-3 fatty acids in each trial supplemented among type 1 diabetic patients.

Our meta-analysis had several strengths that are worth mentioning. First, only RCTs were included. Hence, the bias would be smaller than observational studies due to less confounding. Second, we quantified the association between omega-3 fatty acids and amount of proteinuria and examined it within subgroups. The subgroup analyses allowed the effect of omega-3 fatty acids to be evaluated in specific type of diabetes and follow-up period. In the meanwhile, several limitations of our study should be highlighted. Although, we have the largest sample size, 344 participants were still considered as fairly small number of patients particularly when we performed subgroup analysis. We acknowledged that even after we performed random effects model in our meta-analysis as well as explored for heterogeneity, there are still possible residual confounding such as different background diets of patients or concurrent medications in each trial which were not described. Moreover, different doses and components of omega-3 fatty acids in each trial as well as different control group could lead to heterogeneity and we did not have enough data to perform a dose response meta-analysis. However, EPA and DHA had similar biological actions and properties[43, 44]. Regarding the time of follow-up, median of 24 weeks were relatively short to detect the GFR decline. Furthermore, we had insufficient data on certain clinical parameters regarding duration of diabetes, concurrent medications particularly ACEI/ARB and different methods using to measure urine protein or albumin excretion as an endpoint. Moreover, it was also difficult to conclude whether the effects on proteinuria or other outcomes were caused by EPA or DHA. Furthermore, some biomarkers such as hs-CRP that reflects inflammation were lacking.

In conclusion, the present meta-analysis of 10 RCTs encompassing 344 participants demonstrated that omega-3 fatty acids could ameliorate proteinuria among type 2 DM patients who received omega-3 supplementation for at least 24 weeks without adverse effects on HbA1C, total serum cholesterol and LDL-cholesterol. However, there were no significant difference in change in eGFR between omega-3 fatty acids and placebo group. Clinical trials with more participants and longer time of follow-up should be conducted to better understanding the effects of omega-3 fatty acids on kidney outcomes as well as cardiovascular complications and incidence of stroke among diabetic patients. Markers of oxidative stress, inflammation and urine protein fingerprinting which could reflect severity of glomerular or tubulointerstitial injury should be extensively studied in order to address the potential mechanism of omega-3 fatty acids on delaying proteinuria.

## Supporting information

**S1 Checklist. PRISMA 2009 checklist.**
(DOC)

**S1 Appendix. PubMed search strategy.**
(DOCX)

**S1 Fig. Forrest plots of the included studies assessing HbA1C among diabetic patients.**
(TIF)

**S2 Fig. Forrest plots of the included studies assessing total cholesterol among diabetic patients.**
(TIF)

**S3 Fig. Forrest plots of the included studies assessing HDL cholesterol among diabetic patients.**
(TIF)

**S4 Fig. Forrest plots of the included studies assessing LDL cholesterol among diabetic patients.**
(TIF)

**S5 Fig. Forrest plots of the included studies assessing serum triglyceride among diabetic patients.**
(TIF)

**S6 Fig. Forrest plots of the included studies assessing eGFR among diabetic patients.**
(TIF)

**S1 Table. Summary effects of omega-3 fatty acids on outcomes of interest among diabetic patients (Fixed effects model).**
(DOCX)

## Acknowledgments

We would like to thank Dr. Alessandro Doria at Joslin Diabetes Center and Dr. Murray Mittleman at Harvard T.H. Chan School of Public Health for reviewing and providing comments that greatly improved the manuscript.

## Author Contributions

**Conceptualization:** Api Chewcharat, Pol Chewcharat, Stefania Papatheodorou.

**Data curation:** Api Chewcharat, Anawin Rutirapong.

**Formal analysis:** Api Chewcharat, Stefania Papatheodorou.

**Investigation:** Api Chewcharat, Pol Chewcharat, Anawin Rutirapong.

**Methodology:** Api Chewcharat, Anawin Rutirapong.

**Project administration:** Api Chewcharat.

**Software:** Api Chewcharat.

**Validation:** Pol Chewcharat, Anawin Rutirapong, Stefania Papatheodorou.

**Visualization:** Api Chewcharat, Pol Chewcharat, Anawin Rutirapong, Stefania Papatheodorou.

**Writing – original draft:** Api Chewcharat, Pol Chewcharat.

**Writing – review & editing:** Stefania Papatheodorou.

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
