## [Decision Letter · Decision Letter 0]

10 Oct 2019

PONE-D-19-24289

The Effects of Omega-3 Fatty Acids on Proteinuria among Patients with Diabetes: A Meta-analysis of Randomized Controlled Trials

PLOS ONE

Dear Dr. Chewcharat,

Thank you for submitting your manuscript to PLOS ONE. After careful consideration, we feel that it has merit but does not fully meet PLOS ONE’s publication criteria as it currently stands. Therefore, we invite you to submit a revised version of the manuscript that addresses the points raised during the review process.

We would appreciate receiving your revised manuscript by 9 December 2019. To enhance the reproducibility of your results, we recommend that if applicable you deposit your laboratory protocols in protocols.io, where a protocol can be assigned its own identifier (DOI) such that it can be cited independently in the future. For instructions see: http://journals.plos.org/plosone/s/submission-guidelines#loc-laboratory-protocols

We look forward to receiving your revised manuscript.

Kind regards,

Tomislav Bulum, MD, PhD

Academic Editor

PLOS ONE

Journal Requirements:

- Hu, Jing, Zuoliang Liu, and Hao Zhang. "Omega-3 fatty acid supplementation as an adjunctive therapy in the treatment of chronic kidney disease: a meta-analysis." Clinics 72.1 (2017): 58-64.

- https://www.ahajournals.org/doi/pdf/10.1161/jaha.117.006020?download=true

 The text that needs to be addressed involves some sentences  of the Introduction.

In your revision ensure you cite all your sources (including your own works), and quote or rephrase any duplicated text outside the methods section. Further consideration is dependent on these concerns being addressed.

3. Please include your tables as part of your main manuscript and remove the individual files. Please note that supplementary tables (should remain/ be uploaded) as separate "supporting information" files

4. Please upload a copy of Figure 5, to which you refer in your text. If the figure is no longer to be included as part of the submission please remove all reference to it within the text.

Reviewers' comments:

**Comments to the Author**

1. Is the manuscript technically sound, and do the data support the conclusions?

Reviewer #1: Yes

Reviewer #2: Yes

Reviewer #3: Yes

2. Has the statistical analysis been performed appropriately and rigorously? 

Reviewer #1: Yes

Reviewer #2: Yes

Reviewer #3: Yes

3. Have the authors made all data underlying the findings in their manuscript fully available?

Reviewer #1: Yes

Reviewer #2: Yes

Reviewer #3: Yes

4. Is the manuscript presented in an intelligible fashion and written in standard English?

Reviewer #1: Yes

Reviewer #2: Yes

Reviewer #3: Yes

5. Review Comments to the Author

Reviewer #1: Well done. The author had performed a good meta analysis to answer an important clinical question. Being a meta-analysis , I understand that some of the data eg the use of ACE-I/ARB were not available

Reviewer #2: Title: usually term ‘proteinuria among patients with diabetes’ denotes diabetic nephropathy.

Abstract: No issue except for conclusion. Please refer.

Introduction: The introduction states the gaps in knowledge which justifies a new metaAnalyses .

However, in the 2nd paragraph in reference to the line ‘ However, the effects of omega-3 fatty acids on kidney function still remain controversial particularly in diabetic kidney disease.’ I do not think word 'controversial' is right here. n-3 FAs are perceived to have a role in mediating inflammation, dyslipidemia etc. A more appropriate term would be 'lack of consensus’.

1st Line/Last paragraph/ 1st pp of introduction. The aim of this meta-analysis was to evaluate the benefits of omega-3 fatty acid supplementation in reducing proteinuria in diabetic patients by using all available evidence from the published literature. Isn’t ‘benefits’ pre-conclusive???

Methods: no issue

Results:

2nd line/ 1st paragraphs.. A total of 1277 potentially relevant citations were identified and screened. Seventy citations were evaluated in details, of which 10 trials (refs) with 344 participants…..

Issue- [ref] not stated.

Risk of bias section.. my comment—are the same studies ticking for high risk of bias as per overall risk? Randomization process? Deviation from intended intervention? Missing data?

Subgroup analyses..please quote patient numbers for each group. Also 2nd paragraph/ last line ..should be ‘are’ instead of ‘were’.

Discussion:

There is considerable room for improvement. I would like to detail some points which should be considered:

The major limitation of meta-analyses should be acknowledged. This is it cannot be used to derive explanations for mechanistic pathways or even derive a hypothesis. Further, the construct of this meta-Analyses depends on studies that are poor in design, small in patient numbers and lacking data on more robust biomarkers relating to inflammation [CRP], microalbuminemia status and blood urea levels. There is no point referencing animal studies. There are many studies with CKD population to provide enough inputs as to limitations in these n-3 fatty acid feeding trials.

In addition in reference to Table 1 which describes the selected studies:

There is heterogeneity in treatment based on dose and components of n-3 PUFAs [EPA, DHA or EPA+DHA..an either-and situation]. Further there is also heterogeneity of control treatment. Placebo [not stated] or oleic acid or linoleic acid. The control and treatments DO NOT match. This should be discussed.

Secondly, in these trials the background diets of the patients are not described. As we all know, once a patient is counseled a protein diet [1st line of management to treat proteinuria], then this affects urine protein status. The 2nd aspect is tighter blood glucose control which as its effect on proteinuria, eGFR and HbA1C.

The discussion should raise all the above points.

Conclusion: There is an issue with this statement- ‘omega-3 fatty acids could ameliorate proteinuria among type 2 DM patients who received omega-3 supplementation for at least 24 weeks without adverse effects on HbA1C and serum LDL-cholesterol.’

This conclusion cannot be supported from the evidence reported:

• Overall diabetic patients [NIDDM+IDDM] proteinuria- not significant; eGFR- not significant

• NIDDM- proteinuria- yes ; eGFR- not significant

• IDDM- proteinuria- not significant; eGFR- not significant

• In a meta-regression analysis, the change in proteinuria was not associated with change in GFR (- 0.01 (-0.09, 0.07); p-value = 0.69) and the change in proteinuria was not associated with combined dose of EPA and DHA (0.03 (-0.17, 0.24); p-value = 0.73).

Reviewer #3: Comments:

This manuscript aims to investigate the effects of omega-3 long- chain polyunsaturated fatty acids on proteinuria, eGFR and metabolic biomarkers in diabetic patients. This is a meta-analysis including 10 RCTs with 344 participants, and the authors report that Omega-3 supplementation for 24 weeks or longer could help alleviated proteinuria in patients with type 2 diabetes.

There are some questions should be addressed:

1. Introduction:

(1) Please provide the related references in the paragraph 2.

2. Methods:

(1) Data extraction and quality assessment: how about other serum lipids and glucose control biomarkers, such as the HDL, total cholesterol or fasting glucose?

(2) The results form fixed-effects models should also be presented.

3. Results:

(1) In the flow chart (Figure1), a total of 1277 articles are screened for retrieval, and 179 excluded. However,1089 included in the next stage (missing 9 articles), please check the number carefully.

(2) In table 1, mean age of patients ranged from 33 to 67.4 years old. The duration of follow up spanned from 6 weeks to 52 weeks. It is inconsistent with that in the results section, please check.

(3) Please provide the related tables and figures about the effect of omega-3 fatty acids on eGFR, serum lipids and glucose control.

(4) Why choose 24 weeks as a cut of duration of intervention in subgroup? Please explain. If possible, please provide the results using meta-regression analysis.

(5) Page 13, Paragraph 4: the results were not found in the table 4. Please provide.

(6) Please provide Figure 5.

(7) Each table or figure should be cited in the manuscript. Please check.

(8) Please improve the resolution and clarity of figures.

(9) The authors should provide the mean (SD) of the study outcomes for each treatment group in the figures or tables. It is inappropriate to present MD only.

4. Discussion:

(1) The first paragraph: It is inappropriate to present the result with “this is the first meta-analysis to…among diabetic patients in all aspects.”, because in 2009, Miller et al. conducted a similar meta-analysis.

(2) Please discuss the result that omega-3 fatty acids could help ameliorate serum triglyceride among type 1 DM who received omega-3 supplementation less than 24 weeks.

(3) If possible, to evaluate the optimal dosage of Omega-3 fatty acids for prevention of the study outcomes.

(4) Please further discuss the possible mechanism for effect of omega-3 supplementation on the different diabetes types.

(5) The authors first stated: “the observed effects of omega-3 fatty acids supplementation on proteinuria are not likely the result of blood pressure or renal perfusion effects only because we did not observe simultaneous changes in GFR. Hence, the effect of omega-3 fatty acids in ameliorating proteinuria may be beyond hemodynamic parameters”, while in the followed text, stated: “We found that omega-3 fatty acids did not provide any effects on GFR decline. This could be explained by low sample size as well as short period of follow-up.” Is it reasonable?

5. Please indicate the full names the first time you use the abbreviations in the text.

6. There are some spelling and grammatical errors that should be checked carefully and corrected throughout the manuscript.

---

## [Author Response · Author response to Decision Letter 0]

5 Dec 2019

Responses to Journal:

Thank you for reviewing our manuscript . We edited the whole manuscript according to your format.

Responses to reviewer#1

Reviewer #1: Well done. The author had performed a good meta analysis to answer an important clinical question. Being a meta-analysis, I understand that some of the data e.g. the use of ACE-I/ARB were not available

Response: Thank you for reviewing our manuscript and for highlighting this important point. As we mentioned in the discussion, there was no information about ACE-I/ARB in the primary studies, therefore we were not able to evaluate the effect of this parameter by performing subgroup analysis.

Responses to reviewer#2

Comment#1 Title: usually term ‘proteinuria among patients with diabetes’ denotes diabetic nephropathy.

Response: Thank you for your suggestion. The manuscript’s title has now been changed to “The Effects of Omega-3 Fatty Acids on Diabetic Nephropathy: A Meta-analysis of Randomized Controlled Trials” as suggested

Comment#2 Abstract: No issue except for conclusion. Please refer.

Comment#3 Introduction: The introduction states the gaps in knowledge which justifies a new meta-analyses. However, in the 2nd paragraph in reference to the line ‘ However, the effects of omega-3 fatty acids on kidney function still remain controversial particularly in diabetic kidney disease.’ I do not think word 'controversial' is right here. n-3 FAs are perceived to have a role in mediating inflammation, dyslipidemia etc. A more appropriate term would be 'lack of consensus’.

1st Line/Last paragraph/ 1st pp of introduction. The aim of this meta-analysis was to evaluate the benefits of omega-3 fatty acid supplementation in reducing proteinuria in diabetic patients by using all available evidence from the published literature. Isn’t ‘benefits’ pre-conclusive???

Response: Thank you for your suggestions. We edited these points in the manuscript as you suggested.

Comment#4 Methods: no issue

Comment#5 

Results: 2nd line/ 1st paragraphs. A total of 1277 potentially relevant citations were identified and screened. Seventy citations were evaluated in details, of which 10 trials (refs) with 344 participants…..

Issue- [ref] not stated.

Risk of bias section.. my comment—are the same studies ticking for high risk of bias as per overall risk? Randomization process? Deviation from intended intervention? Missing data?

Subgroup analyses..please quote patient numbers for each group. Also 2nd paragraph/ last line ..should be ‘are’ instead of ‘were’.

Response: 

- Thank you for your suggestion. We edited the references as suggested. 

- Regarding risk of bias, different studies suffered in various domains and Table 2 provides all the necessary information for the specific domains described in the text for each study separately. For overall risk, we combined all of the five elements to summarize the overall risk as recomended from the Revised Cochrane risk-of-bias tool for randomized trials (RoB 2) as referenced. We have now added a more detailed description of the results of the quality assessment in the text as well.

Comment#6

Discussion:

There is considerable room for improvement. I would like to detail some points which should be considered:

The major limitation of meta-analyses should be acknowledged. This is it cannot be used to derive explanations for mechanistic pathways or even derive a hypothesis. Further, the construct of this meta-Analyses depends on studies that are poor in design, small in patient numbers and lacking data on more robust biomarkers relating to inflammation [CRP], microalbuminemia status and blood urea levels. There is no point referencing animal studies. There are many studies with CKD population to provide enough inputs as to limitations in these n-3 fatty acid feeding trials.

In addition in reference to Table 1 which describes the selected studies:

There is heterogeneity in treatment based on dose and components of n-3 PUFAs [EPA, DHA or EPA+DHA..an either-and situation]. Further there is also heterogeneity of control treatment. Placebo [not stated] or oleic acid or linoleic acid. The control and treatments DO NOT match. This should be discussed.

Secondly, in these trials the background diets of the patients are not described. As we all know, once a patient is counseled a protein diet [1st line of management to treat proteinuria], then this affects urine protein status. The 2nd aspect is tighter blood glucose control which as its effect on proteinuria, eGFR and HbA1C.

The discussion should raise all the above points.

Conclusion: There is an issue with this statement- ‘omega-3 fatty acids could ameliorate proteinuria among type 2 DM patients who received omega-3 supplementation for at least 24 weeks without adverse effects on HbA1C and serum LDL-cholesterol.’

This conclusion cannot be supported from the evidence reported:

• Overall diabetic patients [NIDDM+IDDM] proteinuria- not significant; eGFR- not significant

• NIDDM- proteinuria- yes ; eGFR- not significant

• IDDM- proteinuria- not significant; eGFR- not significant

• In a meta-regression analysis, the change in proteinuria was not associated with change in GFR (- 0.01 (-0.09, 0.07); p-value = 0.69) and the change in proteinuria was not associated with combined dose of EPA and DHA (0.03 (-0.17, 0.24); p-value = 0.73).

Response: Thank you for your suggestions. We have reformed the Discussion section accordingly.

We acknowledge the fact that the patient populations in the synthesized studies are different and this heterogeneity led us to apply a random-effects model to analyze the data, incorporating the additional sources of variability between the studies. Regarding background diets, we understand that this is an important confounder in our exposure and outcome association. Unfortunately, there was no available information for this characteristic. However, the randomized nature of the data we are synthesizing provides a reasonable amount of confidence that the comparisons performed are not heavily affected by these factors. We included this lack of information as a limitation in our Discussion section. We are also aware that some RCTs were not optimal regarding study design. We have highlighted this in the Discussion section and also include that as a limitation of our synthesis. 

Regarding the heterogeneity of the treatments and control groups, previous meta-analyses such as Miller et al and Jing et al. performed analyses similar to the current one with the same comparison groups. EPA and DHA both belong to the omega-3 polyunsaturated fatty acids (PUFAs) family and they share common biological properties not only regarding their metabolic and cardiovascular effects but even for anti-cancer effects. They have been evaluated together in the past for various conditions, therefore we considered that it would be reasonable to examine them as equivalent in this meta-analysis. It would be useful to study them separately if we had available evidence, but such data are not available. 

Our conclusion: “omega-3 fatty acids could ameliorate proteinuria among type 2 DM patients who received omega-3 supplementation for at least 24 weeks without adverse effects on HbA1C and serum LDL-cholesterol” takes into account all the points that were raised from the reviewer:

• Overall diabetic patients [NIDDM+IDDM] proteinuria- not significant; eGFR- not significant

• We only refer to type 2 DM, not all diabetics. We refer only to proteinuria, not eGFR or other markers of kidney function.

• NIDDM- proteinuria- yes ; eGFR- not significant

• We do not refer to eGFR in our conclusion 

• IDDM- proteinuria- not significant; eGFR- not significant

• We do not refer to IDDM patient in our conclusion

• In a meta-regression analysis, the change in proteinuria was not associated with change in GFR (- 0.01 (-0.09, 0.07); p-value = 0.69) and the change in proteinuria was not associated with combined dose of EPA and DHA (0.03 (-0.17, 0.24); p-value = 0.73).

• Our conclusion does not refer to the effect of the variables we considered for the exploration of heterogeneity. It only refers to our main analysis.

Responses to reviewer#3

Reviewer #3: Comments:

This manuscript aims to investigate the effects of omega-3 long-chain polyunsaturated fatty acids on proteinuria, eGFR and metabolic biomarkers in diabetic patients. This is a meta-analysis including 10 RCTs with 344 participants, and the authors report that Omega-3 supplementation for 24 weeks or longer could help alleviated proteinuria in patients with type 2 diabetes.

There are some questions should be addressed:

1. Introduction:

(1) Please provide the related references in the paragraph 2.

Response: Thank you for your suggestion. We added the references.

2. Methods:

(1) Data extraction and quality assessment: how about other serum lipids and glucose control biomarkers, such as the HDL, total cholesterol or fasting glucose?

(2) The results form fixed-effects models should also be presented.

Response: Thank you for your suggestion. 

(1) We added as suggested. 

(2) The use of random effects model to synthesize our data was based on an a-priori expectation of substantial between study heterogeneity, mainly due to clinical and methodological reasons. It is expected that the random effects model will yield more conservative results than the fixed effect when tau2 is not equal to zero. Below we demonstrated the results of the fixed-effect model. We added these results of this analysis in the supplementary material. As expected, some outcomes that were not significant in the random effects analysis became significant in the fixed effects model but this is only due to their mathematical properties. We present this table in the supplementary material. 

Outcomes No of study arms No of patients Weighted mean difference/Standardized mean difference* Confidence interval I2 P-value

Proteinuria 11 342 -0.19* (-0.38, 0.01) 0% 0.06

eGFR 6 208 1.54 mL/min/1.73 m2 (-1.40, 4.48) 5.6% 0.31

SBP 10 318 -2.10 mmHg (-4.48, 0.28) 0% 0.08

DBP 10 318 2.10 mmHg (0.57, 3.63) 39.8% 0.007

Triglyceride 10 313 -24.24 mg/dL (-36.40, -12.10) 0% <0.001

TC 6 168 9.07 mg/dl (6.44, 11.71) 80.2% <0.001

HDL-c 6 242 5.97 mg/dl (4.73, 7.22) 82.5% <0.001

LDL-c 6 215 2.29 mg/dL (-2.45, 7.03) 0% 0.34

HbA1C 10 313 -0.42% (-0.60, -0.24) 66.2% <0.001

3. Results:

(1) In the flow chart (Figure1), a total of 1277 articles are screened for retrieval, and 179 excluded. However,1089 included in the next stage (missing 9 articles), please check the number carefully. 

Response: Thank you for your suggestion. It was a typo. 

(2) In table 1, mean age of patients ranged from 33 to 67.4 years old. The duration of follow up spanned from 6 weeks to 52 weeks. It is inconsistent with that in the results section, please check.

Response: Thank you for pointing this out. The data in Table 1 are correct, and we have corrected the numbers in the main text

(3) Please provide the related tables and figures about the effect of omega-3 fatty acids on eGFR, serum lipids and glucose control.

Response: Thank you for your suggestion. The Tables and Figures for eGFR, serum lipids, and glucose control can be found as part of the online Supplement . 

(4) Why choose 24 weeks as a cut of duration of intervention in subgroup? Please explain. If possible, please provide the results using meta-regression analysis.

Response: Thank you for your suggestion. Since there is no formal consensus on the minimum duration of RCTs for this topic and there is large variability in the duration of the studies, we used the median (24 weeks) as a measure of the central tendency of the duration of the studies. This duration (24 weeks) is clinically meaningful as well because the results of supplementation are not expected to be seen in a concise period of time. Therefore, we wanted to differentiate between a true null effect and a null effect because of an inadequate duration of the treatment. We also performed a meta-regression analysis on the duration of follow up as a continuous variable in weeks. The change in proteinuria was not associated with duration of follow-up (-0.01 (-0.02, 0.005); p-value = 0.23). The interpretation of the meta-regression coefficient is the change in proteinuria per 1-week increase in the duration of the treatment which may not be directly interpreted in clinical practice. These results are shown in Table 4. 

(5) Page 13, Paragraph 4: the results were not found in the table 4. Please provide.

(6) Please provide Figure 5.

(7) Each table or figure should be cited in the manuscript. Please check.

(8) Please improve the resolution and clarity of figures.

Response: Thank you for your suggestion. We resolved these issues.

(9) The authors should provide the mean (SD) of the study outcomes for each treatment group in the figures or tables. It is inappropriate to present MD only.

Response: Thank you for your suggestion. In the Forest plots of our analysis, we do report the WMD or SMD of each study along with the 95% CI, which are calculated directly from the SD. In the Tables of the manuscript, because of space limitations, we could not include the individual estimates for all the outcomes. Therefore this information is easily derived from the forest plots. 

4. Discussion:

(1) The first paragraph: It is inappropriate to present the result with “this is the first meta-analysis to…among diabetic patients in all aspects.”, because in 2009, Miller et al. conducted a similar meta-analysis.

Response: Thank you for your suggestion. We downtoned the statement to being the largest meta-analysis among diabetic patients as per your suggestion.

(2) Please discuss the result that omega-3 fatty acids could help ameliorate serum triglyceride among type 1 DM who received omega-3 supplementation less than 24 weeks.

Response: Thank you for pointing this out. After careful review of the numbers, we realized that this was en error at the data entry where the numbers for the control group were placed at the intervention group and vice-versa. With this opportunity, we went back and checked all the data thoroughly, reassuring that there is no other error.

(3) If possible, to evaluate the optimal dosage of Omega-3 fatty acids for the prevention of the study outcomes.

Response: Thank you for your suggestion. Due to lack of available data, we could not perform dose-response analysis to conclude the optimal dose with the least side effects. We added that as a potential limitation to our analysis.

(4) Please further discuss the possible mechanism for effect of omega-3 supplementation on the different diabetes types.

Response: Thank you for your suggestion. We added this part as suggested.

(5) The authors first stated: “the observed effects of omega-3 fatty acids supplementation on proteinuria are not likely the result of blood pressure or renal perfusion effects only because we did not observe simultaneous changes in GFR. Hence, the effect of omega-3 fatty acids in ameliorating proteinuria may be beyond hemodynamic parameters”, while in the followed text, stated: “We found that omega-3 fatty acids did not provide any effects on GFR decline. This could be explained by low sample size as well as short period of follow-up.” Is it reasonable?

Response: Thank you for pointing this out. In this sentence, we tried to provide alternative explanations for the null association between omega-3 fatty acids and GFR, besides the association being indeed null (lack of power or not adequate follow-up). We re-phrased this part of the manuscript to be clearer. 

5. Please indicate the full names the first time you use the abbreviations in the text.

Response: Thank you for your suggestion. We edited as suggested

6. There are some spelling and grammatical errors that should be checked carefully and corrected throughout the manuscript.

Response: Thank you for your suggestion. We went through the manuscript carefully and corrected all the errors.

---

## [Decision Letter · Decision Letter 1]

20 Dec 2019

PONE-D-19-24289R1

The Effects of Omega-3 Fatty Acids on Diabetic Nephropathy: A Meta-analysis of Randomized Controlled Trials

PLOS ONE

Dear Dr. Chewcharat,

Thank you for submitting your revised manuscript to PLOS ONE. After careful consideration, we feel that it has merit but does not fully meet PLOS ONE’s publication criteria as it currently stands. Therefore, we invite you to submit a revised version of the manuscript that addresses the points raised during the review process.

We would appreciate receiving your revised manuscript by 20 January 2020. To enhance the reproducibility of your results, we recommend that if applicable you deposit your laboratory protocols in protocols.io, where a protocol can be assigned its own identifier (DOI) such that it can be cited independently in the future. For instructions see: http://journals.plos.org/plosone/s/submission-guidelines#loc-laboratory-protocols

We look forward to receiving your revised manuscript.

Kind regards,

Tomislav Bulum

Academic Editor

PLOS ONE

Reviewers' comments:

Reviewer's Responses to Questions

**Comments to the Author**

1. If the authors have adequately addressed your comments raised in a previous round of review and you feel that this manuscript is now acceptable for publication, you may indicate that here to bypass the “Comments to the Author” section, enter your conflict of interest statement in the “Confidential to Editor” section, and submit your "Accept" recommendation.

Reviewer #2: All comments have been addressed

Reviewer #3: All comments have been addressed

2. Is the manuscript technically sound, and do the data support the conclusions?

Reviewer #2: Partly

Reviewer #3: Yes

3. Has the statistical analysis been performed appropriately and rigorously? 

Reviewer #2: Yes

Reviewer #3: Yes

4. Have the authors made all data underlying the findings in their manuscript fully available?

Reviewer #2: Yes

Reviewer #3: Yes

5. Is the manuscript presented in an intelligible fashion and written in standard English?

Reviewer #2: Yes

Reviewer #3: Yes

6. Review Comments to the Author

Reviewer #2: Dear Authors,

These details are required:

[1] Table 1 should provide eGFR data

[2] Check reference citation...eg. No.4 is it WE Mitch?? should be.

Reviewer #3: The authors have answered all the questions. However, they still should make a further discussion for the following question in the revised manuscript.

1. The reference 37 is not a meta-analysis, and did not support the related discussion. Please check the order of the references.

7. PLOS authors have the option to publish the peer review history of their article (what does this mean?). If published, this will include your full peer review and any attached files.

Reviewer #2: Yes: Tilakavati Karupaiah

Reviewer #3: No

---

## [Author Response · Author response to Decision Letter 1]

7 Jan 2020

January 6, 2020

Re: Manuscript entitled “The Effects of Omega-3 Fatty Acids on Diabetic Nephropathy: A Meta-analysis of Randomized Controlled Trials ”

 Submission ID: PONE-D-19-24289

Dear Editor,

Thank you for the thoughtful input and review of our manuscript. We believe as a result of this review, our study will have more value for your readers. We revised the manuscript based on the reviewers’ suggestions. We have attached our point by point response. 

Thank you for your time and consideration. If you have any additional questions or comments, please let us know.

Sincerely,

Api Chewcharat, MD, MPH

Department of Epidemiology, Harvard T.H. Chan School of Public Health, Boston, MA 02115, USA 

Email: Api.che@hotmail.com

Responses to reviewer#2

Reviewer #2: Dear Authors,

These details are required:

[1] Table 1 should provide eGFR data

Response: Thank you for your suggestion. We added eGFR data in table 1.

[2] Check reference citation...eg. No.4 is it WE Mitch?? should be.

Response: We appreciated the reviewer’s valuable input. We found the suggested reference very helpful to support our introduction. Hence, we added the reference as suggested. 

Responses to reviewer#3

Reviewer #3: The authors have answered all the questions. However, they still should make a further discussion for the following question in the revised manuscript.

1. The reference 37 is not a meta-analysis, and did not support the related discussion. Please check the order of the references.

Response: Thank you for your comment. To the best of our knowledge, there were no other meta-analyses to support our findings as to why there was only significant reduction in proteinuria only among type 2 diabetes. We postulated that because in type 2 DM, there are higher inflammatory cytokines generated from adipose tissue compared to type 1 DM. Omega-3 fatty acids might help diminish these inflammatory cytokines that eventually leads to lower proteinuria in only type 2 DM. However, we did not have enough information to confirm this hypothesis since the inflammatory markers were not available in many trials that we included. Moreover, it might be explained by the inadequate power to detect the significant difference of proteinuria among type 1 DM because there were only 97 participants in type 1 DM compared to 213 participants for type 2 DM. Future studies are needed to assess whether omega-3 fatty acids could reduce the inflammatory cytokines differently between type 1 and type 2 DM. The following text has been added to the discussion, as suggested. We also removed the reference 37 in the previous manuscript version.

“Our meta-analysis demonstrated only the benefits in delaying proteinuria among type 2 DM patients. This could be explained by a small sample size of type 1 DM patients (213 vs 97). Additionally, the pathophysiology of diabetic nephropathy in type 2 DM and type 1 DM patients is somewhat different. For type 2 DM, proteinuria could be caused by various etiologies including but not limited to insulin resistance, concomitant hypertension and obesity. One of the possible explanations would be that among type 2 diabetes there are pro-inflammatory cytokines generated from abundant adipose tissue as a part of obesity in type 2 diabetes. This inflammatory response leads to proteinuria among diabetic nephropathy. Omega-3 fatty acids help reduce insulin resistance as well as pro-inflammatory responses from adipose tissue. This effect might result in lower proteinuria compared to patients with type 1 diabetes which proteinuria is mainly through polyol, hexosamine, advanced glycation end product and protein kinase C (PKC) pathways (reference 35, 36). Nevertheless, any meta-analyses could not derive explanations for any mechanistic pathways or derive a hypothesis. Hence, future studies designed to examine mechanisms of omega-3 fatty acids on proteinuria or kidney functions are needed as well as to assess the effect of omega-3 fatty acids on inflammatory cytokines among type 1 and type 2 diabetes.” 

For the sentence after that, we would like to raise the point that any meta-analysis could not derive or proof any mechanistic pathway.

We greatly appreciated the editors’ and reviewers’ time and comments to improve our manuscript.

---

## [Editor Report · Decision Letter 2]

14 Jan 2020

The Effects of Omega-3 Fatty Acids on Diabetic Nephropathy: A Meta-analysis of Randomized Controlled Trials

PONE-D-19-24289R2

Dear Dr. Chewcharat,

We are pleased to inform you that your manuscript has been judged scientifically suitable for publication and will be formally accepted for publication once it complies with all outstanding technical requirements.

With kind regards,

Tomislav Bulum

Academic Editor

PLOS ONE

---

## [Editor Report · Acceptance letter]

23 Jan 2020

PONE-D-19-24289R2 

The Effects of Omega-3 Fatty Acids on Diabetic Nephropathy: A Meta-analysis of Randomized Controlled Trials 

Dear Dr. Chewcharat:

I am pleased to inform you that your manuscript has been deemed suitable for publication in PLOS ONE. Congratulations! Your manuscript is now with our production department. 

With kind regards,

on behalf of

Dr. Tomislav Bulum 

Academic Editor

PLOS ONE